# Extracellular space preservation aids the connectomic analysis of neural circuits

**Marta Pallotto[1†], Paul V Watkins[1†], Boma Fubara[1], Joshua H Singer[2], Kevin L Briggman[1,3]***

[1]Circuit Dynamics and Connectivity Unit, National Institute of Neurological Disorders and Stroke, National Institutes of Health, Bethesda, United States; [2]Department of Biology, University of Maryland, College Park, United States; [3]Department of Biomedical Optics, Max Planck Institute for Medical Research, Heidelberg, Germany

**Abstract** Dense connectomic mapping of neuronal circuits is limited by the time and effort required to analyze 3D electron microscopy (EM) datasets. Algorithms designed to automate image segmentation suffer from substantial error rates and require significant manual error correction. Any improvement in segmentation error rates would therefore directly reduce the time required to analyze 3D EM data. We explored preserving extracellular space (ECS) during chemical tissue fixation to improve the ability to segment neurites and to identify synaptic contacts. ECS preserved tissue is easier to segment using machine learning algorithms, leading to significantly reduced error rates. In addition, we observed that electrical synapses are readily identified in ECS preserved tissue. Finally, we determined that antibodies penetrate deep into ECS preserved tissue with only minimal permeabilization, thereby enabling correlated light microscopy (LM) and EM studies. We conclude that preservation of ECS benefits multiple aspects of the connectomic analysis of neural circuits.

*For correspondence: briggmankl@mail.nih.gov

†These authors contributed equally to this work

Competing interests: The authors declare that no competing interests exist.

## Introduction

A number of recent technological advances have automated the collection of serial section electron microscopy (EM) data from the nervous system (*Briggman and Bock 2012*). Complete (connectomic) mapping of synaptic connectivity in these datasets, however, is hampered by the lack of automated analysis methods. The two most important goals for neuronal circuit reconstruction are the reliable reconstruction of neuronal morphologies and the identification of synapses. The delineation of morphologies has proven to be the most difficult step to automate. Current machine learning-based analysis methods enable semi-automated reconstructions (segmentations) of neurons that still require significant human effort to correct (*Helmstaedter et al., 2013*; *Takemura et al., 2013*; *Kim et al., 2014*). Many of the errors encountered during the automated segmentation of neuronal morphologies are related to the dense packing of neurites in neuropil, which frequently leads to the artifactual merging of neighboring neurons (*Jain et al., 2010*). Merging errors are particularly difficult to detect and correct, so most current approaches attempt to reduce merging errors by biasing the output of algorithms to generate over-segmentations of neurons (*Helmstaedter et al., 2013*; *Takemura et al., 2013*; *Kim et al., 2014*). Over-segmentation splits neurons into many small objects that must then be reassembled into complete morphologies, a labor-intensive process. Here, we illustrate that automated segmentation error rates are improved by reducing the packing density of neurites in neuropil.

The preparation of tissue for EM requires finding a compromise among several potential artifacts. One of the most significant, but often underappreciated, artifacts encountered during aldehyde-

**eLife digest** The brain consists of billions of neurons that are connected into many different circuits. Mapping the connections between these neurons could help researchers to understand how the nervous system works. A method commonly used to do so is to preserve samples of brain tissue in chemical fixatives, and then image thin slices of this tissue using powerful microscopes.

As each tissue sample contains many neurons, computer algorithms have been developed to analyze the microscope images and automatically identify the neurons and the connections they make. However, these algorithms often make 'segmentation errors' that researchers need to manually correct: for example, overlapping neurons may be counted as a single neuron, or a neuron may be marked into several segments. Correcting these errors is a time-consuming and tedious task that limits how much of the brain can be currently mapped. Future algorithm improvements will hopefully reduce the number of errors; Pallotto, Watkins et al. explored an alternative approach by making the images themselves easier to analyze using existing algorithms.

The chemicals used to preserve brain tissue often suck out the fluids that fill the spaces between the neurons, causing these 'extracellular spaces' to shrink. Pallotto, Watkins et al. have now developed a method of preserving tissue that maintains more space between the neurons, and used this method to preserve samples of mouse brain with different amounts of extracellular space. Pallotto, Watkins et al. found that the algorithm used to analyze the images of these samples made far fewer segmentation errors on samples that contained more extracellular space. It was also easier to identify the connections between different neurons in these samples. The next challenge will be to extend these methods to preserving extracellular space across whole brains.

based fixation is the loss of extracellular space (ECS). Although this artifact was described decades ago, it has been omitted from major texts describing the ultrastructure of the nervous system (*Peters et al., 1991*). The artifact was first recognized by *Van Harreveld and Malhotra (1967)* and was convincingly demonstrated by comparing the appearance of aldehyde-fixed tissue with that of frozen tissue (*Van Harreveld and Steiner 1970a*; *1970b*; *Harreveld and Fifkova 1975*). The ultra-structure of rapidly frozen tissue revealed appreciable ECS volume fractions of 15–25%, depending on the brain region (*van Harreveld and Khattab 1968*; *Harreveld and Fifkova 1975*; *Korogod et al., 2015*), compared to less than 5% ECS in aldehyde-fixed tissue. The presence of large in vivoECS volume fractions was corroborated by complementary methods including brain conductivity measurements, tracer diffusion studies, and more modern high-pressure freezing experiments (*Rostaing et al., 2004*; *Sykova and Nicholson 2008*; *Korogod et al., 2015*). The primary cause of ECS loss is a net inward flux of ions during tissue fixation, which increases intracellular osmolarity, leading to a redistribution of water into cells and resulting in a swelling of cellular compartments (*Van Harreveld and Malhotra 1967*). Cragg subsequently developed a method to preserve ECS during chemical fixation using a simple ion substitution protocol that replaces extracellular sodium and chloride with a membrane-impermeant molecule such as sucrose prior to fixation (*Cragg 1979*; *1980*), preventing the net inward ion flux that leads to swelling (*Figure 1A*).

Because ECS preservation results in a sparser packing of neurites, we hypothesized that error rates from the automated segmentation of ECS-preserved tissue would be reduced. We tested this hypothesis in the study presented here. Our goal was not to approximate the actual in vivo distribution of ECS. Rather, we reasoned that any manipulation of the ultrastructure that improved image segmentation, including perhaps exaggerating ECS volume fractions beyond physiological ranges, would be beneficial for connectomics so long as the manipulation did not alter synaptic connectivity. By utilizing an ECS-preserving chemical fixation protocol, we were able to titrate the ECS volume fraction in a variety of brain regions. We measured a significant reduction in automated segmentation error rates as the ECS fraction increased, indicating that ECS preservation improves the ability to automatically reconstruct neuronal morphologies from EM data.

We then explored the second important goal of connectomics, the reliable identification of synapses. Electrical synapses are particularly difficult to identify in conventionally prepared (swollen) tissue (*Rash et al., 1998*). We focused on identifying electrical synapses in a known retinal circuit and

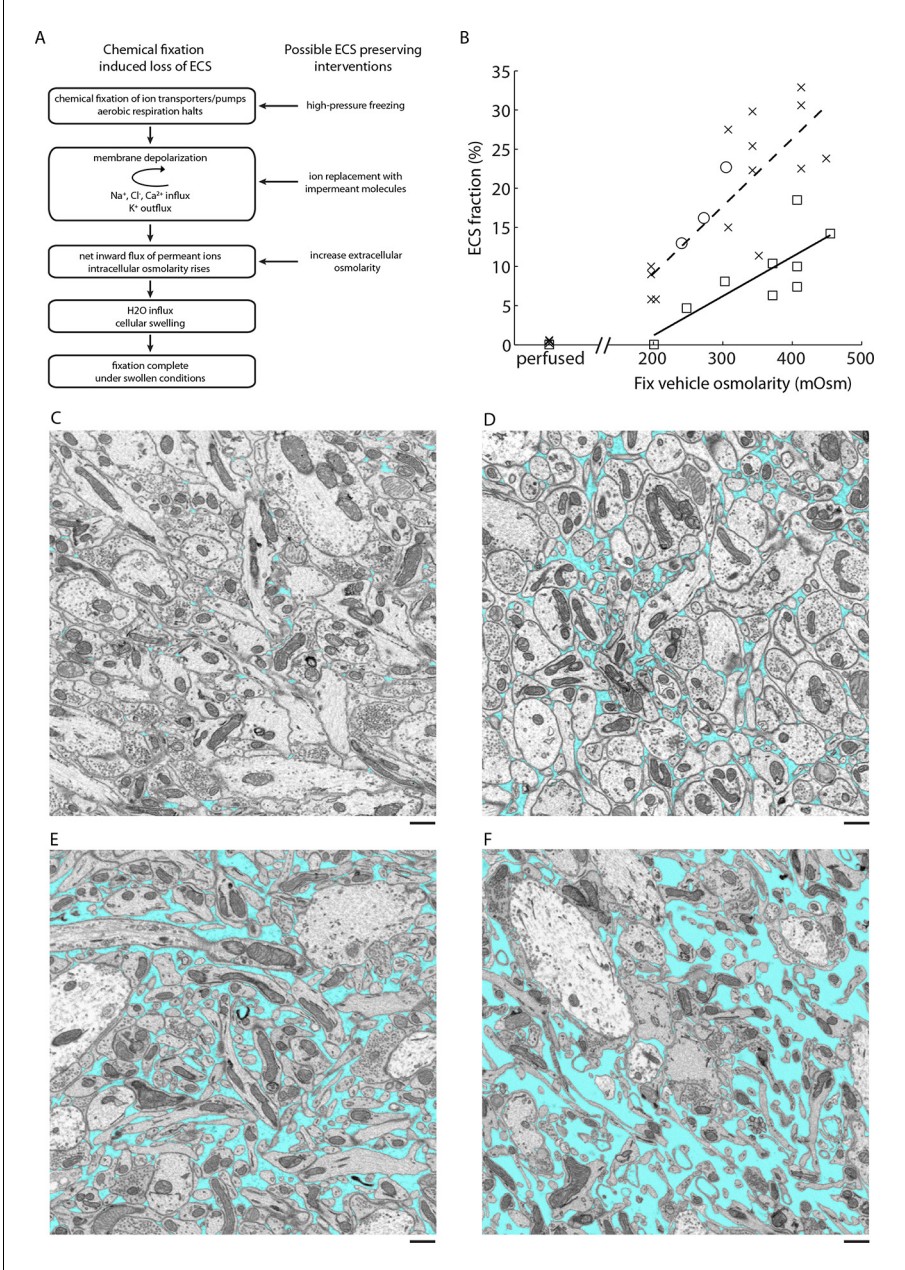

**Figure 1.** ECS preservation in acutely isolated tissues. (**A**) A model of the changes brain tissue undergoes during chemical tissue fixation and possible interventions to preserve ECS. (**B**) ECS fraction of mouse olfactory bulb (x), retina (o), and cortex (□) correlates with increasing fixative vehicle concentration. Dashed line is the linear fit to olfactory bulb data, solid line is the linear fit to cortex data (olfactory bulb: R = 0.80, p = 0.0006; cortex: R = 0.79, p = 0.0113). (**C–F**) EM images from the external plexiform layer of the mouse olfactory bulb from a perfused brain (**C**) or from acute sections fixed with increasing buffer concentrations (**D–F**). ECS fractions are 0.6% (**C**), 5.8% (**D**), 11.3% (**E**), and 23.9% (**F**). Scale bars: 1 μm (**C–F**). ECS: Extracellular space; EM: Electron microscopy.

The following figure supplements are available for figure 1:

**Figure supplement 1.** Change of tissue dimensions and uniformity of ECS preserved sections.

**Figure supplement 2.** Effect of temperature on the preservation of ECS.

found that gap junctions were readily identifiable in ECS-preserved tissue, again due to the increased separation between neurites. Finally, we demonstrate that ECS preservation leads to significant improvements in the diffusion of antibodies into tissue sections with minimal disruption of ultrastructure, an important prerequisite for correlating immunofluorescence with EM. Therefore, we conclude that there are multiple practical benefits to preserving ECS that improve the connectomic analysis of neuronal circuits.

## Results

### Preservation of extracellular space

Cragg described an ECS preservation protocol for chemical immersion fixation that involved briefly bathing acute slices in an unbuffered sucrose solution and then fixing sections in a sucrose/phosphate buffer with glutaraldehyde. We initially applied Cragg's protocol to acute brain sections of the mouse cortex, hippocampus and olfactory bulb and, while ECS was preserved, membranes appeared excessively wrinkled. We therefore developed an alternative strategy to preserve ECS by simply varying the osmolarity of the fixative buffer. Because the loss of ECS during chemical fixation ultimately is due to an imbalance in osmolarity across cellular membranes, we reasoned that adjusting the osmolarity of the fixation buffer would enable us to control the ECS fraction in fixed tissue (*Figure 1A*). It is known that buffer molecules rather than fixative molecules contribute most to the physiological osmotic pressure (*Young 1935*; *Bone and Ryan 1972*). Overall, we noticed smoother membranes using the osmolarity-based approach compared to the sucrose/phosphate protocol.

We observed that the preserved ECS fraction was positively correlated with the osmolarity of the fixation buffer (*Figure 1B,C–F*). For example, a volume fraction of 23.9% in the external plexiform layer (EPL) of the olfactory bulb was observed (*Figure 1F*) when fixed with 2% glutaraldehyde buffered with 175 mM sodium cacodylate buffer (CB, 343 mOsm) compared to 5.8% ECS when fixed with 100 mM CB (204 mOsm, *Figure 1D*). By comparison, fixing tissue by transcardial perfusion, in which the blood–brain barrier remains intact, consistently yielded <1% ECS in the EPL (*Figure 1B,C*). We observed that we could achieve unrealistically large ECS fractions of >25% in the olfactory bulb by raising the buffer osmolarity above 400–450 mOsm, but we began to observe significant membrane ruptures and excessive cellular shrinkage under these conditions. To examine whether overall tissue dimensions changed (either swelled or shrinked) as a function of the preserved ECS fraction, we measured the cross-sectional thickness of 500 µm cortical sections by EM (*Figure 1—figure supplement 1*). We did not observe a correlation between the fixative osmolarity and section thickness (p=0.46, one-way ANOVA), indicating macroscopic tissue dimensions are not affected by osmolarity-based ECS preservation. We also noted that ECS was preserved uniformly across 500-µm-thick sections, the maximum thickness that we tested (*Figure 1—figure supplement 1*). Preservation of ECS was similar whether tissue was sectioned at 4°C or 20°C (*Figure 1—figure supplement 2*), indicating the described protocol is suitable if alterations of neuronal morphology due to cold shock are a concern (*Kirov et al., 2004*; *Bourne et al., 2007*).

We repeated the above experiments for various brain regions including the cerebral cortex and obtained similar results (*Figure 1B*). The only exception was the mouse retina, for which we were unable to preserve ECS by changing the osmolarity of the fixative buffer. Replacing the fixative buffer with unbuffered sucrose, although, restored our ability to preserve ECS in an osmolarity-dependent way (*Figure 1B*). One possible explanation for this is that cellular membranes in the retina are permeable to the cacodylate anion (138 g/mol), but not to sucrose (342 g/mol). This observation indicates that the protocol requires fine-tuning for different brain regions, but we were able to control ECS preservation in every region we have studied.

### Reduction in 2D automated segmentation error rates

Given the ability to predictably control ECS volume fractions, we next investigated whether the increased separation between neighboring neurites in ECS-preserved data would lead to lower error rates in the automated analyses of EM data. Qualitatively, we observed that ECS-preserved data are far easier for human annotators to analyze (e.g. to trace neurons over long distances to create skeleton representations). We therefore asked whether such data also were quantitatively easier to segment by automated machine learning algorithms. We used 2D EM images from the EPL of the

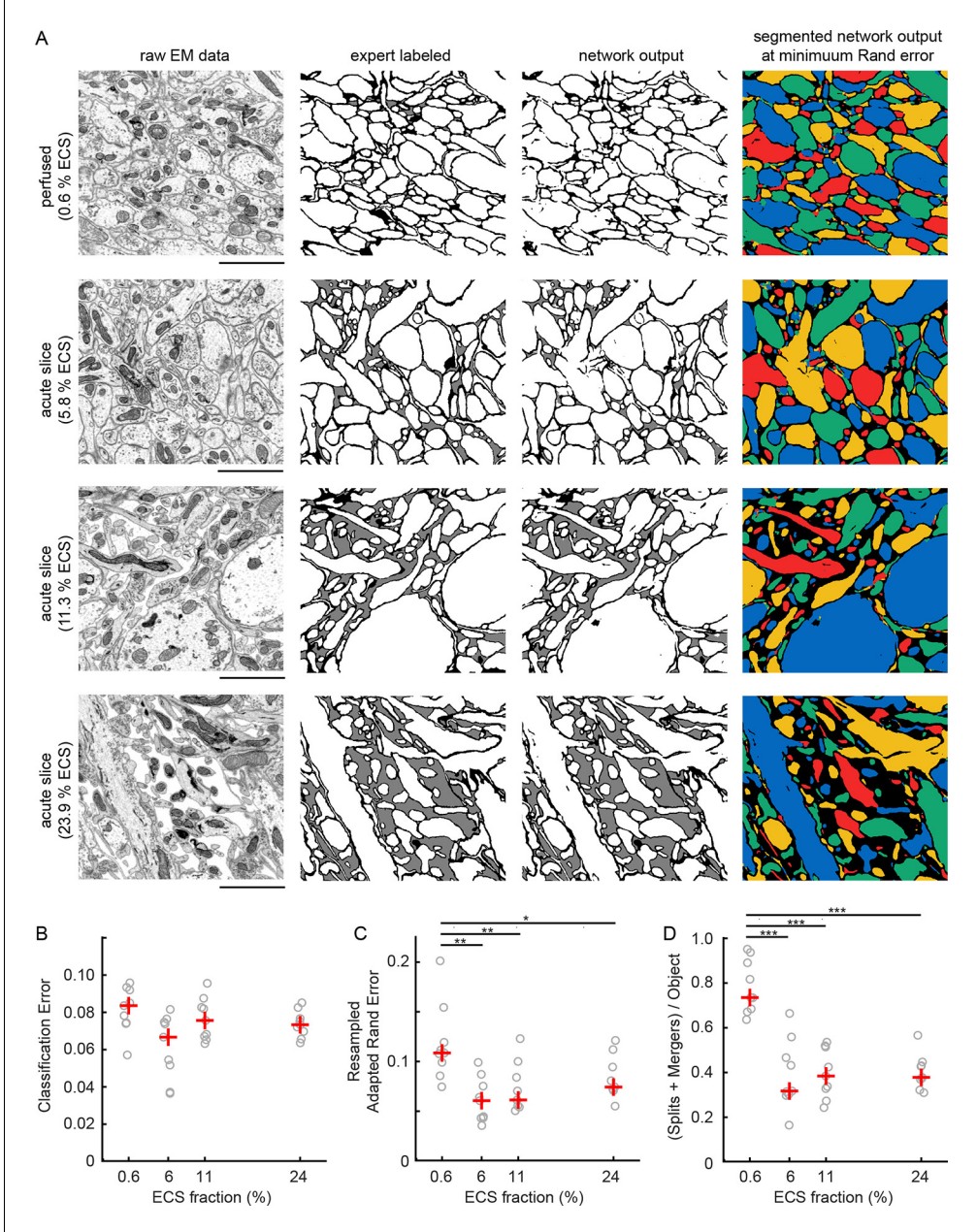

**Figure 2.** Automated 2D segmentation of extracellular space preserved data. (A) Four raw EM images (first column) of varying ECS fractions (0.6, 5.8, 11.3, and 23.9%) were analyzed. The lowest ECS fraction (0.6%) is the perfused preparation while the others are the acute slice preparations. Pixels in each image were annotated (second column) as either intracellular (white), extracellular (gray), or plasma membrane (black). Representative test image pixel classifications (third column) yielded network classification errors of 9.3, 8.1, 6.7, and 7.2% for the four examples shown. Intracellular pixel probabilities were thresholded and segmented (fourth column) to yield minimum Rand errors of 0.11, 0.06, 0.06, and 0.07 for the four examples shown. (B) Pixel classification errors plotted versus ECS fraction from an ninefold cross-validation, median values in red, shows no significant difference between perfused and acute preparations (K-W test: p=0.05). (C) Rand error plotted versus ECS fraction for the ninefold cross-validation, median values in red, shows a significant difference between perfused and acute preparations (K-W test: p<0.01; Wilcoxon rank-sum test: p=0.0017 [5.8%], p=0.0091 [11.3%], p=0.031 [23.9%]). (D) Total number of splits and merger per object in each test image plotted versus ECS fraction for the ninefold cross-validation, median values in red, shows a significant difference between perfused and acute preparations (K-W test: p<0.001; Wilcoxon rank-sum test: p=0.00049 [5.8%], p=0.00035 [11.3%], p=0.00035 [23.9%]). Scale bars: 2 μm. p-Values: *p<0.05, **p<0.01, ***p<0.001. ECS: Extracellular space; EM: Electron microscopy.

The following source data and figure supplement are available for figure 2:

**Source data 1.** Raw image and expert labels of 0.6% ECS data.

*Figure 2 continued*

**Source data 2.** Raw image and expert labels of 5.8% ECS data.
**Source data 3.** Raw image and expert labels of 11.3% ECS data.
**Source data 4.** Raw image and expert labels of 23.9% ECS data.
**Figure supplement 1.** Warping error of automated segmentations.

mouse olfactory bulb as representative samples containing densely packed neuropil. Samples were prepared by perfusion fixation, yielding an ECS fraction of 0.6%, or by acute slice fixation as described above, yielding ECS fractions of 5.8%, 11.3%, and 23.9% (*Figure 2A*). We then manually labeled a portion of these images (*Figure 2A*, second column) and used the labeled data to train convolutional neural networks (CNNs, see 'Materials and methods'). The networks were trained to classify each pixel as belonging either to intracellular space (including cytosol and organelles), to plasma membrane, or to ECS.

We then quantified the number of misclassified pixels, the classification error (*Figure 2A*, third column), on test images. Similar pixel classification errors were observed regardless of ECS fraction (*Figure 2B*), indicating that networks were equally able to learn the statistics of the images. We, however, were interested primarily in the degree to which ECS preservation affected the ability of neural networks to segment actual cells (*Jain et al., 2010*). We therefore joined (segmented) regions of neighboring intracellular pixels and measured segmentation performance using two common segmentation metrics (see 'Materials and methods'). We color-coded segmentations using a four-color map method (*Figure 2A*, fourth column) such that neighboring objects do not share a color. This approach helps to highlight the locations of two predominant error-types: mergers between neurons that should have been segmented as independent objects as well as the splitting of neurons into multiple objects. We observed a significant reduction in segmentation errors that correlated with increasing ECS fraction, regardless of the error metric we used (*Figure 2C, D*, *Figure 2—figure supplement 1*). Therefore, despite sharing similar pixel classification errors, images that contained some degree of ECS were easier to automatically segment. This is because ECS provides more separation between neighboring neurons leading to, for example, fewer mergers between cells.

## Reduction in 3D automated segmentation error rates

It remained possible that while 2D segmentation is improved in ECS data, 3D segmentation would not show a similar improvement perhaps because many ECS-preserved neurite diameters were smaller than their conventionally-fixed counterparts. We therefore collected 3D serial block-face scanning electron microscopy (SBEM, *Denk and Horstmann, 2004*) data from two of the olfactory bulb samples shown in *Figure 2A* (top and bottom rows). We collected 10 x 10 x 12 µm$^3$ volumes from the 0.6% ECS tissue (LowECS) and the 23.9% ECS tissue (HighECS) at a voxel resolution of 9.8 x 9.8 x 25 nm$^3$ (*Figure 3A, B*) (*Pallotto et al., 2015*). We hand-labeled sub-volumes dispersed throughout the volumes as training data and then trained network architectures identical to those used for the 2D automated segmentation in order to classify each voxel. The resulting probability maps were then segmented to create automated labels as a function of threshold (see 'Materials and methods').

To assess segmentation performance across the entire volumes, we densely skeletonized the neurites within each volume; these skeletons served as a ground truth (GT) measure of neurite continuity (*Figure 3A, B*). Notably, the total number of skeletons (LowECS=220, HighECS=217), the total skeletonized path length (LowECS = 3.29 mm, HighECS = 3.40 mm) and the median neurite path length (LowECS = 10.2 [7.6 – 13.3] µm, HighECS = 10.1 [6.6 – 12.7] µm, median [IQR]) were similar between the two datasets. The distributions of neurite path lengths were not significantly different between the two datasets (Kolmogorov–Smirnov test, p=0.91). Together these measurements indicate that the basic statistics of 3D neurite continuity are not altered for the HighECS data compared to the LowECS perfused tissue (*Figure 3—figure supplement 1*).

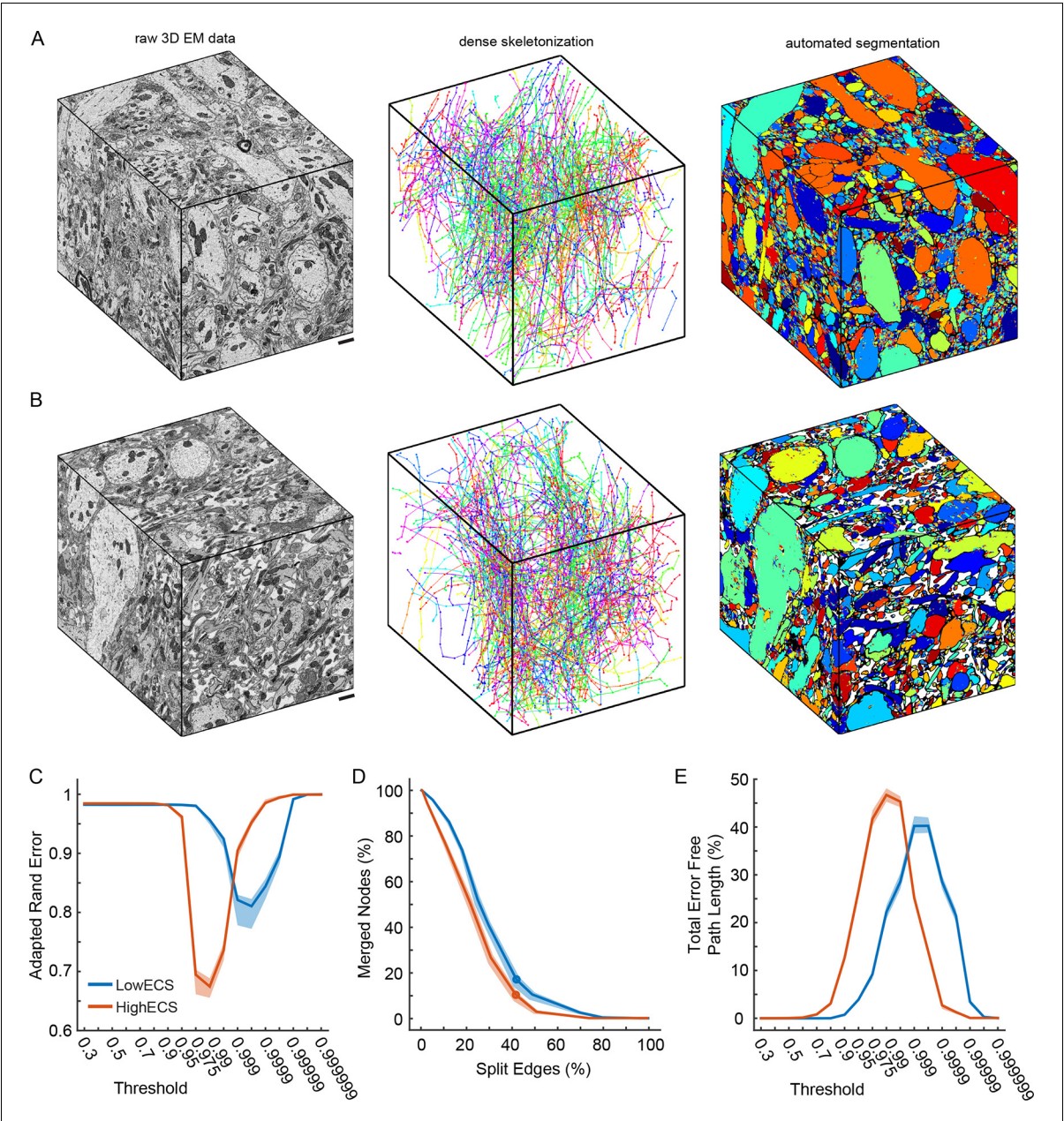

**Figure 3.** Automated 3D segmentation of extracellular space preserved data. (A) A 10 x 10 x 12 μm³ 3D SBEM cube (left) of perfused tissue (LowECS), a dense skeletonization of all neurites in the volume (middle), and an automated segmentation of the volume (right). (B) A 10 x 10 x 12 μm³ 3D SBEM cube (left) of ECS-preserved tissue (HighECS), a dense skeletonization of all neurites in the volume (middle), and an automated segmentation of the volume (right). (C) Adapted Rand error of automated segmentations compared to dense skeletonizations as a function of the segmentation threshold for the HighECS (brown) and LowECS (blue) data volumes. (D) The fraction of merged skeleton nodes versus split skeleton edges as a function of network threshold. Solid points indicate the minimum sum of the error fractions. (E) The total error-free skeleton path length recovered as a function of network threshold. Shaded regions in C–E are the 99% confidence intervals based on a Bernoulli sampling of the data. Scale bars in A, B = 1 μm. SBEM: Serial block-face scanning electron microscopy.

The following source data and figure supplement are available for figure 3:

**Source data 1.** M0027_11 dense skeletonization, Knossos NML file.

**Source data 2.** M0007_33 dense skeletonization, Knossos NML file.

**Source data 3.** M0007_33 training data cubes.

*Figure 3 continued on next page*

*Figure 3 continued*

**Source data 4.** M0027_11 training data cubes.

**Figure supplement 1.** Cumulative distributions for 3D hand-labeled volumes and skeletonizations of the LowECS (blue) and HighECS (brown) data volumes.

We then varied the probability threshold for each network (see 'Materials and methods') and measured the Rand error of the resulting segmentations at the node locations of the dense skeletons. Similar to the 2D results, we observed a significantly lower Rand error for the HighECS dataset (*Figure 3C*). The threshold level that led to the minimum error rate was also lower for the HighECS dataset, indicating that fewer mergers were encountered compared to the LowECS data. We also measured the number of merged skeleton nodes and split skeleton edges as a function of network threshold (*Figure 3D*). The two datasets yielded similar minimum edge split error rates, but the LowECS data achieved a lower node merger error rate. Finally, we measured the total error-free skeleton path length that was recovered as a function of network threshold. We recovered 40% and 47% of the total GT path length for the LowECS and HighECS datasets, respectively (*Figure 3E*). We note that the error metrics were calculated based on a watershed of the network output and that no additional steps to improve the segmentations, such as supervoxel agglomeration (*Jain et al., 2011*; *Nunez-Iglesias et al., 2014*), were applied. In summary, the improvements in automated segmentation that were observed in 2D images were also found in a 3D analysis of ECS preserved compared to perfused tissue.

## Identification of gap junctions in ECS-preserved tissue

We next explored whether ECS preservation would improve the identification of synaptic contacts between neurons. Chemical synapses indicated by clusters of presynaptic vesicles are relatively easy to identify in most brain regions. The identification of electrical synapses (gap junctions), however, remains difficult even in the highest resolution electron micrographs. The number of missed electrical synapses based on thin section images has been estimated to be as high as 75–95% (*Rash et al., 1998*). Unless gap junctions are captured in cross-section in an electron micrograph, electron tomography is usually required to positively identify a gap junction (*Rash et al., 1998*). But performing electron tomography on every putative gap junction in a large EM volume would prohibitively slow data acquisition of even modestly large volumes. We hypothesized that gap junctions would be easier to identify in ECS-preserved tissue given the reduction in incidental (non-synaptic) contacts between cells.

We tested this idea in the well-studied AII amacrine cell circuit of the mammalian retina (*Demb and Singer 2012*). The AII amacrine cell receives ribbon-type chemical synapses from rod bipolar cells and conventional chemical synapses from other amacrine cells. Importantly, it is known that AIIs are coupled electrically to each other as well as to the terminals of ON cone bipolar cells by gap junctions (*Hartveit and Veruki 2012*). We therefore sought to identify gap junctions on a portion of an AII amacrine cell reconstructed from 3D SBEM data.

We volumetrically reconstructed an AII amacrine cell and then annotated all contacts onto a distal portion of its dendritic tree. Contacts were identified as membranes closely apposed (within 50 nm) to the AII's dendrite (n = 171 total contacts). For the majority of these contacts, we observed a clear cleft between the apposed membranes (n = 150 cleft contacts), but, as well, we noted many instances in which we could not observe any cleft (n = 21 tight contacts) (*Figure 4A–D*, see also Supplemental Data stacks). For each contact, we then traced the cell forming the contact through the 3D SBEM volume and classified each cell into four categories: AII amacrine cell, ON cone bipolar cell, presynaptic amacrine cell, or other (see 'Materials and methods'). The 'other' category included a variety of cell types including bipolar, amacrine, ganglion and glial cells that, although making cleft contact with the AII, formed no obvious synapse (e.g. no presynaptic clouds of vesicles were observed). Of the 150 cleft contacts, 13 were attributed to chemical synapses from amacrine cells. The remaining 137 cleft contacts assigned to the 'other' category were deemed to be incidental contacts or ribbon-type synapses (*Figure 4E,F*).

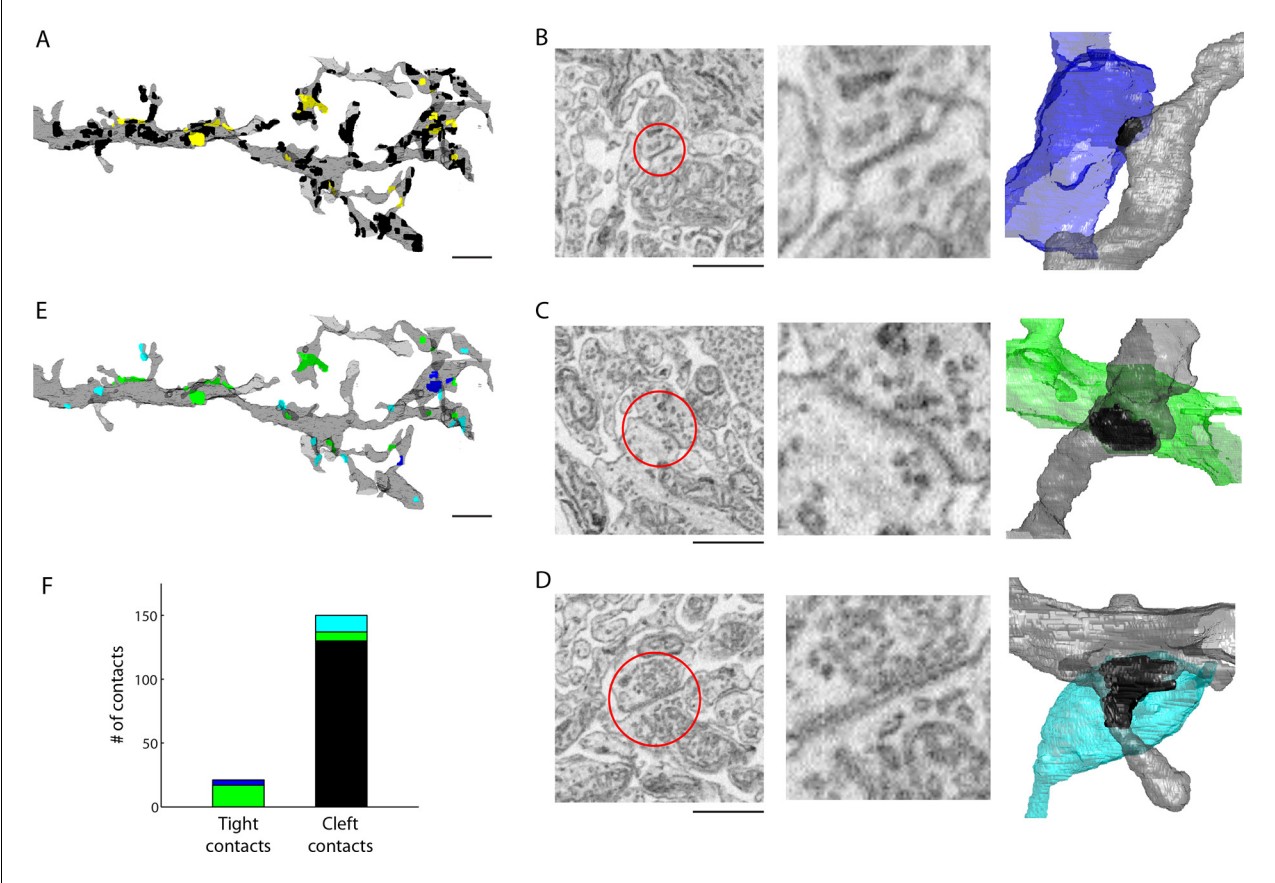

**Figure 4.** Extracellular space aids the identification of gap junctions. (A) Reconstructed dendrites of an AII amacrine cell (gray) and surface renderings of tight (yellow) and cleft (black) contacts. (B-D) Example EM images and volumetric reconstructions of (B) an AII (gray) to AII (blue) tight contact, (C) an AII (gray) to cone bipolar cell (green) tight contact (C), and (D) a cleft contact identified as a chemical synapse between a presynaptic amacrine cell (cyan) and the AII cell (gray). (E) Tight contact surface renderings color-coded by the identity of the synaptic partner: AII amacrine cell (blue), cone bipolar cell (green), presynaptic amacrine cell (cyan). (F) Summary of the number of contacts color-coded by the cell type of the contacting cell: AII (blue), cone bipolar (green), presynaptic amacrine cell (cyan), other (black). Scale bars: 2 µm (A,E), 1 µm (B–D). EM: Electron microscopy.

The following source data and figure supplement are available for figure 4:

**Source data 1.** Example image stack of an A2 to A2 tight contact.

**Source data 2.** Example image stack of a cone bipolar to A2 tight contact.

**Source data 3.** Example image stack of a chemical synapse to A2 cleft contact.

**Figure supplement 1.** Simple measures of contact geometry do not predict gap junctions.

We then asked whether the tight contacts were actually indicative of gap junctions (electrical synapses) onto the AII amacrine cell. Of the 21 tight contacts we annotated, 4 were formed with a neighboring AII amacrine cell and 17 were formed with ON cone bipolar cell terminals (*Figure 4E, F*). Thus, in every instance, the tight contact was formed with a cell type known to be electrically coupled to the AII amacrine cell indicating that tight contacts in ECS preserved data are consistent with gap junctions. Moreover, AII-AII gap junctions were found in sublamina 5 of the IPL, and AII-ON cone bipolar gap junctions were found in sublaminae 3 and 4, as previously reported (*Strettoia et al., 1992*). We noted that the gap junctions were not identifiable based on contact geometry measurements, such as surface area, alone (*Figure 4—figure supplement 1*). The

preservation of ECS therefore enabled the identification of gap junctions due to the relative scarcity of contacts amongst cells and the simple segregation of contacts into tight versus cleft categories.

## Improved diffusion depths of antibodies

Finally, we explored whether ECS preservation could improve the ability to correlate fluorescent immuno-labeling of proteins with EM ultrastructure. A long-standing problem with preparing brain tissue for immunohistochemical observation is the incomplete diffusion of antibodies into large tissue volumes. Antibodies are large proteins and therefore strong permeabilization of tissue is typically required for labeling thick tissue sections. Often, the degree of permeabilization required to label thick samples destroys the fine tissue ultrastructure. We hypothesized that the preservation of ECS would allow antibodies to penetrate deeper into tissue and would, therefore, require only minimal permeabilization. We tested this idea in the mouse retina using antibodies against the vesicular acetylcholine (VAChT) and GABA (VGAT) transporters, two antibodies that generate distinct labeling patterns in the retina. Following a fixation and minimal permeabilization procotol that we developed in order to preserve ECS and decent tissue ultrastructure (see 'Materials and methods'), we incubated 200-µm-thick retinal sections with primary and secondary antibodies for relatively short durations (9 hr each) and measured the labelling depth. We observed deeper penetration of both antibodies in ECS-preserved retina (*Figure 5A–C*). VAChT penetrated 28.1 ± 2.4 µm in ECS-preserved samples versus 6.8 ± 0.5 µm in control samples; VGAT penetration was 16.9 ± 1.0 µm versus 6.1 ± 0.4 µm for ECS-preserved and control samples, respectively. The labeling patterns were consistent with the known expression patterns of the two transporters in the retina (*Koulen 1997*; *Johnson et al., 2003*). We processed these same pieces of retina for EM and confirmed that the

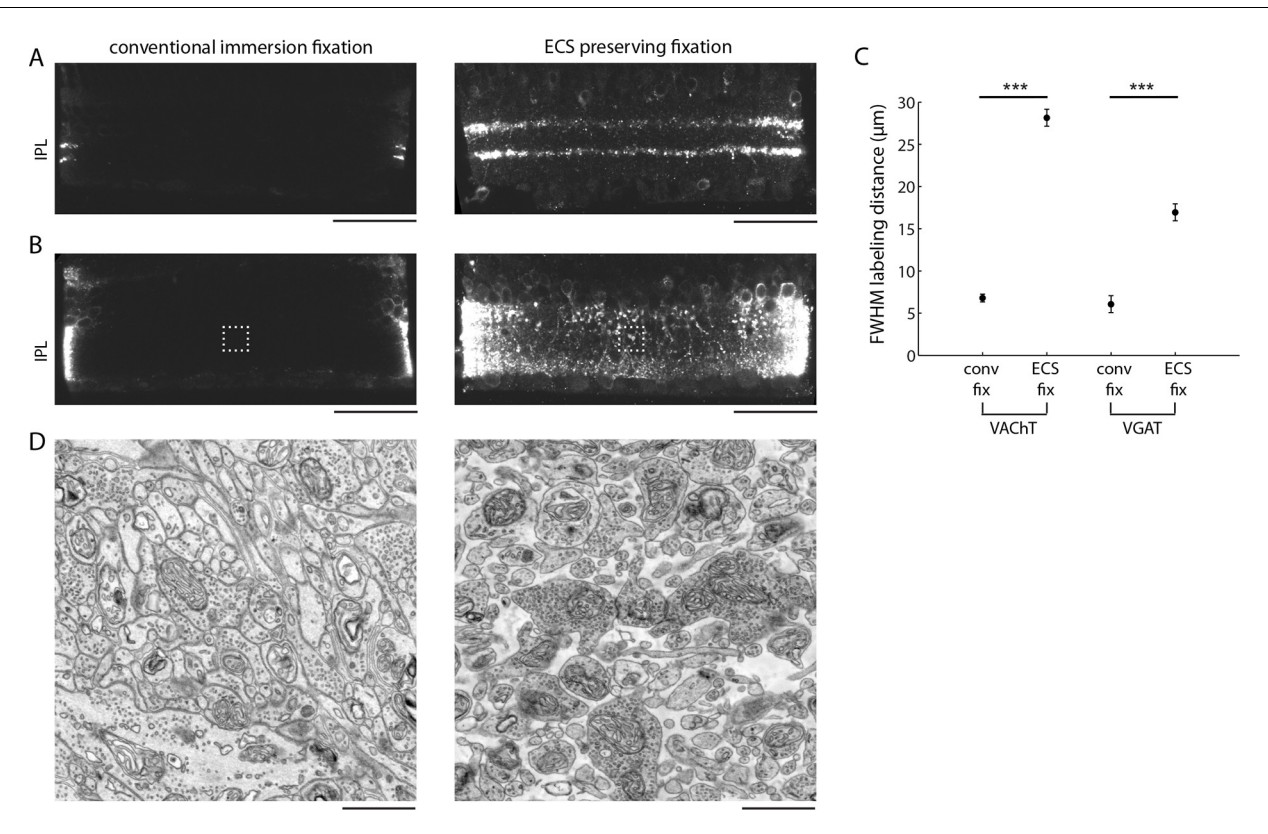

**Figure 5.** Extracellular space preservation increases access to antibodies. (**A**) Flat-mounted retinas were immunolabeled for vACHT with and without ECS preservation. Retinas were then cross-sectioned and confocal images were taken to directly visualize penetration depth. (**B**) Same as **A**, but immunolabeled for vGAT. (**C**) Fullwidth half-maximal penetration distances (mean +/- SEM) for retinas with and without ECS preservation (n = 20 retina slices from two animals, unpaired Student's t-test, vACHT: ***p<0.0001, vGAT: ***p<0.0001). (**D**) EM images collected from the regions of interest in panel **B** indicated by dotted rectangles. Scale bars: 50 µm (**A,B**), 1 µm (**D**).

penetration depth correlated with the ECS fraction (*Figure 5D*, 0.2% vs. 19.2% ECS fraction). Furthermore, the ultrastructure of the weakly permeabilized tissue was well preserved (*Figure 5D*).

## Discussion

The major bottleneck for automating connectivity mapping in large neuronal circuits is no longer data acquisition (*Briggman and Bock 2012*), but rather data analysis. Manual data analysis allows targeted reconstructions of small population of neurons (*Briggman et al., 2011*) but is unfeasible for dense reconstructions. Automating image segmentation is therefore necessary for any reasonable scalability; trained human annotators, however, will still be required to correct the results of automated image processing algorithms (*Helmstaedter et al., 2013*; *Takemura et al., 2013*; *Kim et al., 2014*). Therefore, any reduction in the error rates of automated segmentations of large EM datasets will directly reduce the required human effort to correct, or 'proofread', errors.

Our findings indicate that preserving ECS aids the connectomic analysis of neuronal circuits in multiple ways. The loss of ECS during the conventional chemical fixation of brain tissue leads to densely packed neurites and tightly apposed plasma membranes that complicate automated analyses of EM data. We reasoned that preserving some degree of ECS would improve automated image segmentation methods due to the increased separation of non-synaptic membrane appositions. We emphasize that our goal was not to preserve in vivo-like ECS or neuronal volumes. Indeed, freezing tissue is likely the only way to approximate in vivo-like morphologies (*Korogod et al., 2015*). Our goal was rather to measure image segmentation performance as a function of increasing ECS volume fractions. In other words, because changes in ECS are unavoidable during the chemical fixation of brain tissue, why not adjust the artifact to be more beneficial to us from a data analysis standpoint? Indeed, it was possible that actually exceeding in vivo ECS volume fractions would lead to the best segmentation performance. On the contrary, we found that while preserving ECS improved segmentation performance, the performance did not continue to improve with increasing ECS fractions (*Figure 2C,D*). One of the reasons for the diminishing returns on segmentation performance is the shrinkage of cells that accompanies increasing ECS fractions which makes segmenting thin processes, particularly glial sheets, more error prone. There, therefore, appears to be an optimal ECS fraction from the standpoint of image segmentation that balances the increased separation between neurites with the shrinkage of cellular compartments. Modest ECS fractions of 6% already significantly improve segmentation performance compared to fractions of <1% (*Figure 2C,D*), typical of standard perfusion fixation protocols.

Most current EM segmentation strategies rely on generating over-segmentations (neurons are split into many objects) to avoid accidental mergers between neighboring neurons. These over-segmented objects then are joined together algorithmically (*Jain et al., 2011*; *Nunez-Iglesias et al., 2013*; *2014*) and/or by human annotators (*Takemura et al., 2013*; *Kim et al., 2014*). The segmentation of ECS-preserved data results in fewer mergers, leading to better initial segmentations, and therefore should reduce the human effort required to generate accurate reconstructions. We modified a state-of-the-art CNN originally designed for natural image classification (*Krizhevsky et al., 2012*) to analyze EM data. The CNNs we trained achieved similar pixel classification error rates regardless of the ECS fraction of a sample (*Figure 2B*), but segmentation errors were reduced substantially by increasing ECS fractions (*Figure 2C,D* and *Figure 3*). In other words, pixel classification errors have a more detrimental impact on segmentation when cells are tightly packed versus when they are separated by ECS. This basic observation should hold regardless of the particular analysis approach used (*Rigoll et al., 2008*; *Jurrus et al., 2010*; *Turaga et al., 2010*; *Berning et al., 2015*). Our analysis showed that the improved segmentation performance in 2D images also extends to 3D segmentation (*Figure 3*).

In order to prepare tissue with the ECS volume fractions we desired, we modified an existing protocol for ECS preservation in chemically fixed tissues. The modification we made involved simply changing the osmolarity of the buffer used during fixation. This approach, however, was not successful in the mouse retina; we had to rather change to unbuffered sucrose fixation to preserve ECS in the retina. We have successfully preserved ECS uniformly across 500-μm-thick acute brain slices, the thickest slices we tested. High-pressure freezing is an alternative to chemical fixation that also preserves ECS (*Rostaing et al., 2004*; *Sykova and Nicholson 2008*). The maximum slice thickness that is adequately preserved using this technique, however, is at most 200 μm: too thin to contain many

neuronal circuits. The chemical preservation protocol we used is also far less expensive to implement than high-pressure freezing and is simple to evaluate in different brain regions and species.

A 500-µm section is a sufficiently large volume to contain many of the local neuronal circuits in the brain, such as a cortical column. Because improvements in data acquisition speeds eventually will allow larger volumes to be acquired, extending uniform ECS preservation to, for example, a whole mouse brain is an important goal. A previous study described a transcardial perfusion-based approach to preserve ECS throughout the brain involving transient opening of the blood-brain barrier and ion replacement (*Cragg 1980*), but the uniformity of the ECS fraction across the brain was not evaluated. Given the benefits we observed, further exploration and validation of this whole-brain ECS preservation method is warranted.

The unambiguous identification of gap junctions is a long-standing problem in the analysis of brain circuits by EM. When ECS is lost, conventional transmission EM with pixel resolutions of a few nm can resolve gap junctions only when they are oriented such that they are captured in cross-section; electron tomography is required to resolve gap junctions oriented obliquely to the image plane (*Rash et al., 1998*). Performing electron tomography on each putative gap junction in a large tissue volume is impractical. We therefore determined whether ECS preservation aided gap junction detection at a moderate pixel resolution (10 nm) using SBEM (*Denk and Horstmann, 2004* ) data. We found that subsets of all contacts onto an AII amacrine cell in the mouse retina are tight contacts with no visible cleft. We identified all these tight contacts as being formed with cells known to be coupled electrically to AIIs (*Figure 4*). Although we performed this analysis manually, we anticipate a relatively straightforward method for automating the detection of tight contacts between cells. Along the same lines, the automated identification of chemical synapses may also be aided simply because there are fewer incidental contacts between cells in ECS-preserved data. Importantly, the analysis of the AII amacrine cell circuitry demonstrates that the known synaptic connectivity is present in ECS-preserved tissue.

Correlating fluorescence immunohistochemistry with EM data can add significant functional information to purely structural descriptions of circuits. However, there is usually a tradeoff between the degree of permeabilization required for antibody penetration and the quality of ultrastructural preservation. The preservation of ECS retains large diffusion pathways, allowing deeper penetration of large antibody proteins into tissue (*Figure 5*). The increased access to antibodies allowed for the use of a minimal permeabilization pre-embedding protocol that preserves satisfactory tissue ultrastructure for subsequent EM examination of antibody-labeled tissue.

The ECS-preserved EM images of the brain we report here appear substantially different from the vast majority of published EM data over the last 50 years. Practitioners of rapid freezing-based tissue preservation, however, have appreciated the loss of ECS as a major artifact in conventional EM tissue processing for decades. We wish to, in particular, note the pioneering work of Anthonie Van Harreveld who first elucidated this artifact in EM and whose careful analyses directly led to the ECS-preserving interventions that we employed. The relatively minor changes to existing tissue fixation protocols that preserve ECS ultimately benefit several aspects of the connectomic analysis of neuronal circuits.

## Materials and methods

### Tissue fixation protocols

We fixed and examined tissue from a variety of brain regions of C57BL/6 mice, aged 9-12 weeks in accordance with NIH animal ethics guidelines.

For ECS preservation in acute slices, mice were deeply anesthetized using a mixture of ketamine hydrochloride (100 mg/kg IP) and xylazine hydrochloride (10 mg/kg IP) or isoflurane (Forane, Baxter, Deerfield, IL) and then swiftly decapitated. We cut 300–500-µm slices on a vibratome (Leica) according to the procedure of (*Bischofberger et al., 2006*). Slices were cut at 4°C or at 20°C in ACSF containing (in mM): 124 NaCl, 3 KCl, 1.3 $MgSO_4.7H_2O$, 84 $NaHCO_3$, 1.25 $NaH_2PO_4.H_2O$, 20 glucose, 2 $CaCl_2.2H_2O$ and equilibrated with a 95% $O_2$ – 5% $CO_2$ gas mixture. Slices were then transferred into the fixative solution at 20°C, containing 2% glutaraldehyde (GA, Electron Microscopy Sciences, Hatfield, PA) buffered (pH 7.4) with one of the following sodium-cacodylate buffer (CB, Sigma-Aldrich, St. Louis, MO) concentrations (in mM): 100, 150, 175, or 200. Slices were fixed for

8 hr, rinsed with CB buffer of the same concentration used for fixation for 4 hr and then stained for EM.

For perfusion-fixed tissue, we followed the rapid perfusion approach of (*Tao-Cheng et al., 2007*). Mice were transcardially perfused with 2% GA, 2% paraformaldehyde (PFA, Electron Microscopy Sciences), in 150 mM CB buffer (pH 7.4). The brain was extracted and postfixed for 8 hr in the same fixative solution. Subsequently, the tissue was rinsed in 150 mM CB buffer for 4 hr. Tissue sections from the perfused brains were cut on a vibratome (Leica, Germany) at 300–500 µm and stained for EM.

For isolated retinas, retinas were isolated from the eye cup in Ames solution (Sigma-Aldrich), flat mounted on filter paper and then incubated for 5 min at 20°C in a modified Ames solution. The modified Ames solution was prepared by diluting Ames 1:1 with distilled water and then adding 5% sucrose. We then transferred retinas into a fixative at 20°C containing 2% GA in unbuffered sucrose (for ECS preservation) or 2% GA buffered with 150 mM CB (pH 7.4). Retinas were fixed for 1 hr, rinsed with 150 mM CB buffer and stained for EM. Retina tissue samples were taken from an eccentricity approximately halfway between the optic disk and the peripheral edge of the retina.

## EM tissue staining

We used the ROTO (reduced osmium-thiocarbohydrazide-osmium) *en bloc* staining protocol suitable for SBEM previously described (*Briggman et al., 2011*). Briefly, samples were stained in a solution containing 1% osmium tetroxide, 1.5% potassium ferrocyanide, and 150 mM CB for 2 hr at room temperature. The osmium stain was amplified with 1% aqueous thiocarbohydrazide (1 hr at 50°C), and then 2% aqueous osmium tetroxide (1 hr at room temperature). The samples were then stained with 2% aqueous uranyl acetate for 12 hr at room temperature and lead aspartate for 2–12 hr at room temperature. Samples were embedded in Epon resin.

## ECS quantification from 2D images

All 2D EM images were acquired from ultrathin (50–100 nm) sections mounted on copper TEM grids in a scanning electron microscope with a field-emission cathode (Nova NanoSEM 450, FEI Company, Netherlands) using a solid-state back-scattered electron detector. Incident beam energies were 2.0–2.5 kV and pixel resolution was typically 9.8 nm. For quantification of ECS in 2D, we randomly selected 9.8 x 9.8 µm$^2$ regions from EM images of dense neuropil and manually labeled ECS pixels. Labeling was performed blinded to the fixation conditions. We intentionally avoided annotating regions containing cells bodies or blood vessels that would distort ECS fraction estimates due to their large volumes. For olfactory bulb data, we collected images from the neuropil of the EPL. For retina data, we imaged the neuropil of the inner plexiform layer. For cerebral cortex, we imaged neuropil from layers 2/3. The ECS percentage was measured as the fraction of labeled ECS pixels in the annotated region.

## Antibody labeling

Flat-mounted retinas were incubated in the modified ACSF solution for 5 min at 20°C and then fixed for 1 hr with 2% PFA + 0.01% GA in either 7.5% sucrose (for ECS preservation) or 150 mM CB (pH 7.4). Retinas were then rinsed 3 x 15 min in 150 mM CB at room temperature, embedded in 3% agarose prepared in 150 mM CB and vibratome sectioned into 200µm slices. The slices were cut at an eccentricity approximately halfway between the optic disk and the peripheral edge of the retina. Slices were rinsed in 50 mM glycine in 150 mM CB for 30 min, and in a 300 mOsm PB-BSA washing buffer containing 120 mM phosphate buffer (pH 7.4), 0.5% BSA (Sigma-Aldrich), and 0.05% sodium azide (Sigma-Aldrich) for 2x 10 min. They were then transferred to a blocking solution containing 120 mM PB (pH 7.4), 1% BSA, 10% normal donkey serum (NDS, Abcam, United Kingdom), 0.5% Tween 20 (Sigma-Aldrich), and 0.05% sodium azide for 1 hr at 20°C. Primary antibody staining was performed on free floating agitated slices for 9 hr at 4°C with antibodies targeting either the vesicular GABA transporter (Synaptic Systems, Germany, cat no. 131003, rabbit anti-VGAT) or the vesicular acetylcholine transporter (Synaptic Systems, cat no. 139103, rabbit anti-VAChT) at dilutions of 1:250 and 1:300, respectively, in the blocking solution except with 3% NDS (instead of 10% NDS). Retinas were then rinsed in PB-BSA for 2 hr at 4°C and then stained with a donkey anti-rabbit fluorescent secondary antibody, DyLight 650 (Abcam), at 1:300 dilution in the blocking solution (with 3% NDS)

for 9 hr at 4°C. Retinas were then rinsed in PB-BSA 2x 10 min and transferred to 150 mM CB. The slices were then re-embedded in 3% agarose in 150 mM CB and cut in half to assay the degree of antibody penetration. Retinal cross-sections from the cut surface were imaged on a confocal microscope (Carl Zeiss AG, Germany) with a 633-nm laser and a LP650 emission filter. All images were acquired under identical confocal imaging parameters. The antibody penetration distance was measured by averaging the signal across the depth of the inner plexiform layer and quantifying the full-width half-maximum (FWHM) distance of the staining profiles from the two edges. The FWHM distances from both edges were averaged to yield one data point per tissue slice. Following confocal imaging, all retinas fixed again in 150 mM CB + 2% GA and were subsequently stained for EM using the method above.

## SBEM data collection

For the 3D segmentation analysis, we collected two volumes from samples obtained from the EPL of the mouse olfactory bulb containing 0.6% (dataset M0027_11, LowECS) or 23.9% (dataset M0007_33, HighECS) 2D ECS fractions. Data were collected using a custom serial block-face microtome designed by K.L. Briggman. The specimens were cut out of the flat-embedding blocks and re-embedded in Epon Hard on aluminium stubs. The samples were then trimmed to a block face of ~200 μm wide and ~300 μm long. The samples were imaged in a scanning electron microscope with a field-emission cathode (NanoSEM 450, FEI). Back-scattered electrons were detected using a concentric segmented back-scatter detector. The incident electron beam had an energy of 2.4 keV and a current of ~200 pA. Images were acquired with a pixel dwell time of 2 μs and size of 9.8 nm × 9.8 nm which corresponds to a dose of about 42 electrons/nm$^2$. Imaging was performed at high vacuum, with the sides of the blocks sputter coated with a 100-nm-thick layer of gold. The section thickness was set to 25 nm. Five hundred and twelve consecutive block faces were imaged from each sample, resulting in aligned data volumes of 4096 × 3536 × 512 voxels (corresponding to an approximate spatial volume of 40 × 35 × 12.8 μm$^3$). Then, 10 × 10 × 12 μm$^3$ regions were selected from the two volumes for the 3D segmentation analysis.

For the gap junction analysis of the retina, a 73.2 x 27.6 x 57.6 μm$^3$ (6100 x 2300 x 2300 voxels) block of ECS-preserved mouse retina (dataset k0731, 17.4% ECS) was acquired by SBEM at a resolution of 12 x 12 x 25 nm$^3$ using a microtome designed by W. Denk. The dataset spanned the inner plexiform layer including the ganglion cell layer and a portion of the inner nuclear layer. The retina was cut out of the flat-embedding block and re-embedded in Epon Hard on aluminium stubs. The retina was then trimmed to a block face ~200 μm wide and ~200 μm long. The samples were imaged in a scanning electron microscope with a field-emission cathode (UltraPlus, Carl Zeiss). Back-scattered electrons were detected using a custom-designed detector based on a special silicon diode (AXUV, International Radiation Detectors, Torrance, CA) combined with a custom-built current amplifier. The incident electron beam had an energy of 1.4 keV and a current of ~1 nA. Images were acquired with a pixel dwell time of 0.45 μs and size of 12 nm × 12 nm, which corresponds to a dose of ~31 electrons/nm$^2$. Imaging was performed at high vacuum, with the sides of the blocks sputter coated with a 100-nm-thick layer of gold. The section thickness was set to 25 nm.

All data sets were split into cubes (128 × 128 × 128 voxels) for viewing in KNOSSOS (www.knossostool.org, *Helmstaedter, et al., 2011*).

## Gap junction analysis

An AII amacrine cell was initially identified by tracing a cell that received a ribbon type synapse from a rod bipolar cell. A portion of this AII cell was volumetrically reconstructed manually using ITK-Snap (www.itksnap.org, [*Yushkevich et al., 2006*]). Contacts onto the surface of the AII from neighboring cells were manually annotated and each contact was classified as either a tight contact, based on the absence of any discernable gap between the two membranes, or a cleft contact for membranes that came within ~50 nm of each other. K.L.B and J.H.S. then independently skeletonized the cells forming these contacts using Knossos (www.knossostool.org). Cells were skeletonized until each cell's identity could be unambiguously determined as either: an AII amacrine cell (based on visualizing connectivity with rod bipolar cell terminals), a cone bipolar cell (based on presence of ribbon synapses and the absence of rod bipolar cell connectivity), a presynaptic amacrine cell (based on presence of a conventional chemical synapse at the contact), or 'other'. The 'other' category

included ganglion cell dendrites, amacrine cell dendrites, rod or cone bipolar cell axons not forming synaptic junctions, and Muller glial cells. None of the cells in the 'other' category formed a chemical synaptic contact onto the AII amacrine cell.

## Automated segmentation: training and testing data

For 2D segmentation comparisons, test and training images were obtained from EM images from the EPL of the mouse olfactory bulb containing 0.6%, 5.8%, 11.3%, or 23.9% ECS. A portion of each of these images (1920 x 1728 pixels, 18.8 μm x 16.9 μm) was hand-labeled using ITK-Snap, and each pixel was classified as either: membrane (MEM), ECS or intracellular space (ICS). GT integer labels were obtained from these hand-labeled images by running connected components (four connectivity) on ICS pixels only. The labeled region was subdivided into nine sub-images of size 640 x 576 pixels to perform a ninefold cross-validation of neural network performance. For each EM image, nine independent CNNs were trained on eight of the sub-images and then tested on the remaining sub-image.

For 3D segmentation comparisons, training cubes were obtained from the M0027_11 and M007_33 data volumes. We densely hand-labeled voxels from six sub-volumes (each 128 x 128 x 128 voxels, 1.25 x 1.25 x 3.20 μm³) contained within the data volumes. Sub-volumes were labeled so that ICS voxels from distinct neurites had different integer labels and so that ECS voxels were all labeled with the same integer label. Manual labels were automatically corrected by, in order: (1) removing very small foreground (ICS and ECS) and background (MEM) components of less than 5 pixels in xy, xz, and yz (orthogonal) image slices (eight connectivity); (2) removing adjacency between ICS components with different labels in orthogonal image slices (eight connectivity) by setting pixels on each side of any adjacencies to MEM; (3) finding ICS components with the same label value that were not connected in 3D and assigning different label values to unconnected components (six connectivity); and (4) removing very small 3D ICS labels that contained less than 27 voxels.

## Automated segmentation: dense skeletonization

The M0027_11 and M0007_33 data volumes were densely skeletonized using Knossos for comparison with full 3D segmentations. Only neurons were skeletonized, glial processes (identified based on cellular morphology) and ECS were not skeletonized.

## Automated segmentation: neural network training

We used a CNN with a parallelized GPU implementation (*Krizhevsky et al., 2012*; *Zeiler and Fergus 2013*) modified from (https://github.com/akrizhevsky/cuda-convnet2). Networks were trained on 3GB Nvidia GTX 780Ti and 6GB GTX TITAN Black graphics cards (EVGA). The CNNs processed 2D images to learn labels for both 2D and 3D EM data. Training images were sampled from randomized pixels in the EM data such that the network was trained to learn, in parallel, the identity of all the pixels (ICS, ECS, or MEM) in a 16 x 16 pixel patch located in the center of 64 x 64 pixel input images (*Jain et al., 2007*; *Ciresan et al., 2012*). We trained seven-stage convolutional networks containing (1) an input convolutional layer with 96 kernels of size 7, response normalization across 31 feature maps, and max pool subsampling of size 3 with stride of 2 (overlapped pooling resulting in an overall downsampling factor of 2); (2) a convolutional layer with 256 kernels of size 5 and the same normalization and subsampling as the first layer; (3,4) two convolutional layers with 384 kernels of size 3; (5) a convolutional layer with 256 kernels of size 3 and the same subsampling as the first layer; (6,7) two fully connected hidden layers of size 4096 each with a dropout rate of 0.5 (*Hinton et al., 2012*); and (8) an output layer of size 768 (16x16 output pixels x 3 pixel identities). Each output represents the conditional probability that a particular output pixel belongs to one of ICS, ECS or MEM (*Bishop 1995*). Convolutional layer units used the ReLU nonlinearity (*Krizhevsky et al., 2012*) while fully connected hidden layer units used a linear activation function. The output layer used the logistic nonlinearity and gradients were calculated using a cross-entropy error function for multiple independent attributes (*Bishop 1995*). Average gradients were calculated and back-propagated for each mini-batch size of 128 image examples. Weights in the fully connected and output layers were updated with a weight decay of 0.02 and all weights and biases used a weight momentum term of 0.9 (*Krizhevsky et al., 2012*). Randomized training images were augmented using simple transformations of the input image, which included all combinations of

transposes and reflections for a total of eight possible augmentations (in two dimensions) (*Ciresan et al., 2012*).

For 2D image segmentations, nine networks were trained for each ECS dataset using a ninefold cross-validation described above. For 3D volume segmentations, five networks were trained for each volume using all six sub-volumes of training data. Three networks were trained using images taken from the xy planes of each volume (based on the original orientation of the EM data). Two networks were trained by reslicing the original volumes to create xz and yz oriented images. The final voxel probabilities used to create EM segmentations were a weighted average of the five networks, with the probabilities from the xz and yz networks weighted at 0.5 relative to the xy networks. The randomized presentation of images to the network was balanced so that the center pixel ICS, ECS, or MEM identities were presented with equal probability. Because this procedure changes the prior probabilities and does not guarantee the same equal balancing for all 256 (16 x 16) output pixels, the output probabilities of test images were reweighted using Bayes' rule (*Vucetic and Obradovic 2001*). The test probabilities (the target prior) represented the actual frequency of ICS, ECS and MEM voxels over all training data. Training frequencies used in the reweighting procedure were measured by tallying the count of ICS, ECS, and MEM pixels that were actually presented to each of the 256 outputs during training.

Networks were trained using 0.8 million images per training epoch. Learning rates for all weights were initialized at 0.00005 and for all biases at 0.0001. Different randomizations of the training data were presented during each epoch. The learning rate decayed exponentially but discretely 16 times per epoch with a time constant of 5 epochs. All networks were run for 10 training epochs.

## Automated segmentation: segmentation

Network output probabilities for each output were exported for test voxels and the output was classified as either MEM, ECS, or ICS based on the maximum probability (winner-take-all).

To create segmented labels, we used a customized watershed procedure that relied on iteratively running connected components to find local peaks within the ICS and ECS CNN probability outputs for each pixel. ICS and ECS probabilities were set to zero if they were not the winning probability, and then probabilities were binarized at increasing thresholds. For 2D segmentations the thresholds were 0.3, 0.4, 0.5; then from 0.5 to 0.99 in steps of 0.01; and then 0.995, 0.999, 0.9995, 0.9999, 0.99995, 0.99999, 0.999995, and 0.999999. For 3D segmentations thresholds were 0.3 to 0.9 in steps of 0.1; then 0.95, 0.975, 0.99; and then 0.995, 0.999, 0.9995, 0.9999, 0.99995, 0.99999, 0.999995, and 0.999999. At each binarization step, connected components (four connectivity for 2D, six connectivity for 3D) were created for all pixels above the threshold. Any components that fell below a minimum size threshold parameter ($T_{min}$) of 64 pixels for 2D or 256 voxels for 3D were accumulated in a binary mask. Tests for the sensitivity of this parameter revealed that it had less than 1% impact on all metrics measured across the ranges tested ($T_{min}$ = 8, 16, 32, 64, 128, 256). This step essentially was a method for detecting local peaks in the probabilities. Moving from low to high thresholds, connected components was run on the logical OR of the accumulated binary mask and the remaining binarized pixels above the current threshold level. These components were then dilated toward filling in the surrounding region of voxels with the matching winner-take-all identity using a topological warping technique that does not allow components to become split or merged (*Kong and Rosenfeld 1989*; *Legland et al., 2011*). Any remaining ICS or ECS winner-take-all voxels were merged to the nearest (Euclidean distance) ICS or ECS component, respectively. For display of 2D segmentations, we used four colors to label the different components so that no neighboring components were the same color (*Appel and Haken 1977*).

## Automated segmentation: 2D metrics

Pixel categorization error was calculated as the fraction of correctly labeled pixel identities from the CNNs (winner-take-all max probability out of ICS, ECS, MEM) compared to GT test images. To compare components of automated segmentations and GT segmentations, we used the Adapted Rand Error (ARE) and warping error to compare ICS components only.

The ARE was calculated as 1 minus the F-score of the precision and recall of the Rand Index (*Hubert and Arabie 1985*; *Jain et al., 2010*). In order to account for significantly different GT component sizes (pixels per component), which can bias the ARE when comparing different ECS

datasets, we applied a Bernoulli sampling procedure that sampled (without replacement) GT ICS components and pixels within those components. We used the minimum number of components (n = 71) across the 36 possible sub-images (four ECS fractions x nine sub-images) as the expected number of components selected on each sampling iteration. A portion of pixels within each of the sampled components was then selected. We used the minimum number of pixels per component (n = 20) as the expected number of pixels selected from each component. The ARE was then calculated using only the selected pixels for a particular sampling iteration. We used the median ARE of 1000 samples as the final metric for this procedure.

We also compared 2D segmentations using the warping error (*Jain et al., 2010*). Difference pixels were classified by the type of topologic error that they represented, and only split and merger pixels were counted. Connected split and merger pixels (i.e., belonging to the same components being split or merged) were counted as a single split or merger by running connected components (eight connectivity) on pixel errors. We divided the number of splits and mergers by the number of GT components in each test image to fairly compare between datasets (yielding splits per GT component and mergers per GT component).

## Automated segmentation: 3D metrics

For 3D segmentations, we compared automated segmentations with the dense skeletonization for each data volume. Using the skeletons as GT, we calculated: (1) a skeleton node-based ARE; (2) the fraction of split edges and merged nodes and (3) the total error-free path length (TEFPL).

Overlap between skeletons and segmented 3D supervoxels was calculated as a confusion matrix, where rows correspond to skeletons and columns to supervoxels. Each entry in the matrix is the number of nodes on a given skeleton that are contained within a particular supervoxel. The sum total of all entries is therefore equal to the total number of skeleton nodes (n=5130 LowECS, n=5364 HighECS). The ARE was then calculated based on this confusion matrix using the same calculation as that used for pixel-wise 2D segmentation. Bernoulli sampling was not required in this case as the number of nodes per skeleton was not statistically different between the datasets (*Figure 3—figure supplement 1A*). Any nodes that overlapped with membrane areas of the segmentations were represented in the confusion matrix as belonging to a single membrane supervoxel. Both ICS and ECS supervoxels were used in creating the confusion matrix. Overall, only a very few number of skeleton nodes fell into either membrane (n=8, LowECS; n=12, HighECS) or ECS supervoxel locations (n=0, LowECS; n=1, HighECS).

We also used the supervoxel/skeleton confusion matrix to calculate the fraction of merged nodes. Summing along the rows of the logical confusion matrix (true at locations with at least one node) creates a marginal that indicates which supervoxels overlap with more than one skeleton. All nodes that are within these supervoxels were then counted as merged nodes. The number of merged nodes can then range from zero to the total number of nodes (if all nodes were part of a single supervoxel), which we divided by to get the fraction of merged nodes. To calculate the fraction of split edges, we iterated over the edges of skeletons. An edge was split if both nodes belonging to the edge were located in different supervoxels. If either node was located in a membrane area, the edge was also considered as being split. The number of split edges can then range from zero to the total number of edges (if all supervoxels only contained a single node), which we divided by to get fraction of split edges. To calculate TEFPL, an edge was error free if it was not split and if neither of its nodes was a merged node. We summed the path lengths of all error free edges to get TEFPL, which was expressed as a percentage of the total path length in each volume.

## Statistical analyses

All statistical analyses were performed using GraphPad Prism (GraphPad Software) or Matlab (The Mathworks). For comparison between two groups of equal size, the non-parametric Kruskal–Wallis test was applied to assess whether there was any significance between medians, then the Wilcoxon rank-sum test was used for pairwise comparisons in the cases where the Kruskal–Wallis test was significant (*Figure 2B–D*). For comparison between two groups of equal size, unpaired t-tests were used (*Figure 5C*). For correlation coefficients, the Student's t-test was used with n-2 degrees of freedom (*Figure 1B*). Confidence intervals for 3D skeleton-based error metrics were calculated

using a Bernoulli sampling with an expected value of 95% of the minimum number of skeletons (n = 206).

## Acknowledgements

This work was supported by the National Institute of Neurological Disorders and Stroke Intramural Research Program. JHS was supported by NIH Grant EY017836. We thank W Denk for supporting the initial stages of this research, many helpful discussions and encouragement; A Merrihew and P Ghorbani for volumetric tracing of the SBEM retina dataset; J Shlens for helpful discussion regarding machine learning algorithms; and G Murphy, J Diamond and W Denk for constructive comments on the manuscript.

## Additional information

### Funding

| Funder | Grant reference number | Author |
|--------|------------------------|--------|
| National Institute of Neurological Disorders and Stroke | Intramural Research Program (NS003133) | Marta Pallotto<br>Paul V Watkins<br>Boma Fubara<br>Kevin L Briggman |
| National Eye Institute | EY017836 | Joshua H Singer |
| Pew Charitable Trusts | Pew Scholars Program in the Biomedical Sciences | Kevin L Briggman |
| Max-Planck-Gesellschaft | | Kevin L Briggman |

The funders had no role in study design, data collection and interpretation, or the decision to submit the work for publication.

### Author contributions

MP, KLB, Conception and design, Acquisition of data, Analysis and interpretation of data, Drafting or revising the article; PVW, Conception and design, Analysis and interpretation of data, Drafting or revising the article; BF, Acquisition of data, Drafting or revising the article; JHS, Analysis and interpretation of data, Drafting or revising the article

### Author ORCIDs

Marta Pallotto, http://orcid.org/0000-0001-7694-0398

### Ethics

Animal experimentation: We fixed and examined tissue from a variety of brain regions of C57BL/6 mice, aged 9 to 12 weeks in accordance with NIH animal ethics guidelines. All of the animals were handled according to an approved institutional animal care and use committee (IACUC) protocol. The protocol was approved by the NINDS Animal Care and Use Committee (#1340-15).

## Additional files

### Major datasets

The following datasets were generated:

| Author(s) | Year | Dataset title | Dataset URL | Database, license, and accessibility information |
|-----------|------|---------------|-------------|--------------------------------------------------|
| Pallotto M, Watkins PV, Fubara B, Singer JH, Briggman KL | 2015 | Extracellular space preservation aids the connectomic analysis of neural circuits | http://dx.doi.org/10.5061/dryad.36h28 | Available at Dryad Digital Repository under a CC0 Public Domain Dedication |

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
