## [Decision Letter]

Thank you for submitting your work entitled "Extracellular space preservation aids the connectomic analysis of neural circuits" for peer review at *eLife*. Your submission has been favorably evaluated by Eve Marder (Senior Editor), a Reviewing Editor, and two reviewers.

The reviewers have discussed the reviews with one another and the Reviewing editor has drafted this decision to help you prepare a revised submission.

Summary:

The paper by Pallotto et al. addresses a significant problem in the field of serial electron microscopy applied to connectomics: the inability of current segmentation algorithms to accurately and quickly trace the finest neuronal wires and their connections. The approach of this paper is very interesting and relatively novel since the investigators are focused on improving sample preparation for segmentation algorithms rather than what the majority of the field is focused on, building better algorithms. The issue is related to the shrinkage of the extracellular space associated with the normal aldehyde fixation and processing procedures used for EM. As has long been known (from Van Harreveld's work in the 1950's and 1960's), the ECM is compressed during fixation and subsequent processing, as compartments like dendrites and astroglia swell a little at the expense of the ECM, which is compressed. Strategies to avoid this involve difficult to implement methods such as high pressure freezing and freeze substitution, (the methods introduced by Van Harreveld). This extreme method of stopping action by vitrification is not easy to use or apply to brains without also being concerned with the effects of anoxia on the way to the snap-freezing step. Anoxia itself results in similar differential changes in compartment sizes, at the expense of the ECM space. High osmolality and pressure in perfusions have been tried but results are very inconsistent. This work chose to investigate the utility of a work around using stabilized slice preparations, thus avoiding the concerns above, but introducing the obvious issue of the preparation having been ripped from the brain and then stabilized to be as good as a slice is in faithfully representing the in vivo physiology and anatomy. Since it is well known that significant synaptic remodeling occurs during slice stabilization (at least for the hippocampus where it has been studied carefully) the results here must be considered in the context of the preparation having already endured some likelihood of synaptic and other morphological changes. Nevertheless, the work shows a practical method for swelling extracellular space to make automated tracing more feasible for connectomics work. However, care should be taken to make sure the community does not conclude that this makes the spaces "as they were" in nature. There are a number of other serious concerns that need to be addressed in the revised version.

Essential revisions:

1) The authors have not convincingly demonstrated that preservation of the extracellular space will actually substantially improve the automatic segmentation process. The authors only examined 2D segmentation and even within the 2D segmentation results, it is not clear that a certain fraction of ECS maximizes segmentation accuracy with the various metrics used to evaluate segmentation accuracy (see Figure 2). Even more problematic is the lack of 3D automatic segmentation results. This is worrisome since there seem to be an abundance of very small neuronal processes (due to the inevitable 'shrinking' of cells caused by ECS preservation relative to standard EM datasets.) This has been well known in the efforts of the current connectomics community for preservation of the ECS and has remained a source of errors for tracing cellular processes in EM datasets. Examples of small processes can be seen in Figure 1 and it is not clear whether they can be accurately traced over 3D either manually or by automatic segmentation. Without 3D segmentation results of these small processes, it remains possible that ECS preservation improves 2D segmentation but could actually worsen 3D segmentation and not improve our ability to trace the finest neuronal processes. Thus 3D segmentation results are important for demonstrating that this study has made a significant advance.

2) Although the authors clearly demonstrate that ECS preservation can improve human annotation of putative electrical contacts, it is not clear from the study the false positive and negative rates for this detection. We agree with the authors that putative electrical synapses seem to occur in their dataset between cells known to have these connections, but it is difficult to evaluate whether it is possible to get an accurate number of these connections. Given that a 'goal' of dense circuit reconstruction is to move past qualitative descriptions of the type of connectivity between neurons (i.e. they are electrically or synaptically coupled) to quantitating numbers of connections, data about false positive and negative rates would make this result more compelling.

3) The argument that this approach is going to be valuable for connectomics should rely on the assumption (or better a demonstration) that connections are not altered during acute slice preparation. While it may be true that large-scale connectivity could not change in a few minutes during slice prep, however it is a shame that the authors do not reference the work of Kristin Harris' group showing that spines are lost on neurons in slices during cold shock (Kirov et al., Neuroscience 2004). There was also a follow-up paper showing that this can be largely avoided by preparing slices at room temp (Bourne et al., Neuropharmacology 2007). In the current manuscript, the authors are slicing before fixation at ice-temp. They do not state so clearly in the Materials and methods, but one has to assume from the Materials and methods description here that the slices then went directly into ice-cold fixative. So there is no warm recovery period and one would expect that there would be a severe loss of spines and other changes in microtubules, known to be cold sensitive. True, the synapses are probably still there, but they would appear to be onto dendrites instead of spines. This could make it difficult to distinguish excitatory and inhibitory synapses, for one thing. It means that this approach would not be well suited for use in any study focused on neuronal morphology, rather than just finding connections (with the concern about artifacts of slice preps as above). The authors should ensure that they further emphasize these potential limitations of their approach.

4) In the improved diffusion depths of antibodies experiment, authors should provide the negative control data in both fixation conditions. Did authors harvest the retina pieces in the same location? Sample size is critical for the antibody penetration because flat mounted retina has different thickness in the central, middle and peripheral portion. Additionally, slice images of XZ, YZ should be added in the Figure 4 to provide convincing idea.

---

## [Author Response]

*Essential revisions: 1) The authors have not convincingly demonstrated that preservation of the extracellular space will actually substantially improve the automatic segmentation process. The authors only examined 2D segmentation and even within the 2D segmentation results, it is not clear that a certain fraction of ECS maximizes segmentation accuracy with the various metrics used to evaluate segmentation accuracy (see Figure 2). Even more problematic is the lack of 3D automatic segmentation results. This is worrisome since there seem to be an abundance of very small neuronal processes (due to the inevitable 'shrinking' of cells caused by ECS preservation relative to standard EM datasets.) This has been well known in the efforts of the current connectomics community for preservation of the ECS and has remained a source of errors for tracing cellular processes in EM datasets. Examples of small processes can be seen in Figure 1 and it is not clear whether they can be accurately traced over 3D either manually or by automatic segmentation. Without 3D segmentation results of these small processes, it remains possible that ECS preservation improves 2D segmentation but could actually worsen 3D segmentation and not improve our ability to trace the finest neuronal processes. Thus 3D segmentation results are important for demonstrating that this study has made a significant advance.*

Regarding 2D segmentation results, only Figure 2 report segmentation performance based on the Rand and Warping error metrics. These two metrics show a statistically significant improvement in segmentation performance for all ECS volume fractions compared to the perfused tissue. Figure 2 shows the pixel classification error that is not a metric for segmentation performance. This error metric is sensitive to minor differences in how, for example, pixels are labeled near membranes. For example, if a neural network consistently provides a slightly eroded segmentation relative to ground truth labels, the pixel classification error will be high even though this may not change the resulting segmentation. These issues are discussed in: V. Jain, H. S. Seung, and S. C. Turaga. Machines that learn to segment images: a crucial technology for connectomics.Curr Opin Neurobiol. 20, 653-66 (2010). This reference is cited in the paper (subsection “Reduction in 2D automated segmentation error rates”).

We have added 3D data and an analysis of 3D segmentation performance to the revised manuscript. These results are presented in the new Figure 3 and discussed in the text under the heading ‘Reduction in 3D automated segmentation error rates’. We collected 10 x 10 x 12 µm^3^ cubes of data from the samples shown in Figure 2, top row (LowECS) and bottom row (HighECS). We trained identical network architectures on sub-volumes from the two 3D datasets. We then densely skeletonized all neurites in the two datasets. Importantly the basic statistics of the skeletons derived from both datasets, including total path length and path length per neurite, were very similar. This serves as another indication that the underlying topology of neurite continuity is not altered by the ECS preservation protocol.

We then compared the segmentations obtained by varying the threshold on the network output against the skeletons and measured the Rand error, the fraction of merged nodes versus split edges, and total error free skeleton path length. We again observed an improvement in all three of these metrics for the ECS preserved data compared to perfused data, consistent with the 2D results.

*2) Although the authors clearly demonstrate that ECS preservation can improve human annotation of putative electrical contacts, it is not clear from the study the false positive and negative rates for this detection. We agree with the authors that putative electrical synapses seem to occur in their dataset between cells known to have these connections, but it is difficult to evaluate whether it is possible to get an accurate number of these connections. Given that a 'goal' of dense circuit reconstruction is to move past qualitative descriptions of the type of connectivity between neurons (i.e. they are electrically or synaptically coupled) to quantitating numbers of connections, data about false positive and negative rates would make this result more compelling.*

We agree that deriving false negative rates for detection remains difficult. We refer the reviewers to the estimate from Rash, J. E., T. Yasumura, et al. (1998). "Ultrastructure, histological distribution, and freeze-fracture immunocytochemistry of gap junctions in rat brain and spinal cord.” These authors estimate that the false negative rate for detecting electrical synapses even in high-resolution TEM sections is 75-95% (subsection “Identification of gap junctions in ECS preserved tissue”), an astonishingly high number . Block-face SEM data is typically of a lower lateral resolution than TEM sections (but substantially better Z resolution), 10 x 10 x 25 nm^3^ voxels are state-of-the-art, which may lead to an even worse false negative rate. Our view is that the preservation of ECS at least makes it more likely to identify tight contacts that are oriented at arbitrary angles in 3D. That is, ECS preservation shifts detecting electrical synapses from ‘nearly undetectable’ to ‘detectable’ in SBEM data. We view this as a significant advance. Because we do not have a ground truth estimate of the number of gap junctions between AII-AII cells or between AII-CBC cells, we cannot estimate how many we may have missed. We could explicitly label for gap junctions, but that would require the collection of another large dataset and we feel this is beyond the scope of the paper.

On the other hand, an estimate of false positive rates can be derived from the data presented. Any tight contact that was formed with a cell known to not electrically coupled to AII amacrine cells is a false positive. In our sample, 0 of 21 tight contacts were formed with cells that do not couple to the AII. Therefore, given our sample, our current false positive rate is 0. We note two important details. First, the judgement of whether a contact was tight or cleft was made blind to the identity of the postsynaptic process. It is therefore not possible that we unintentionally biased our results towards cells known to couple to the AII. Second, the distribution of tight contacts along the axis of the inner plexiform layer (IPL) is in agreement with previously described positions of gap junctions. AII-CBC gap junctions are observed at the most distal tips of AIIs and AII-AII gap junctions are observed more proximal to the soma. Our results also show this pattern, further supporting the finding that we did not annotate false positives in our sample.

*3) The argument that this approach is going to be valuable for connectomics should rely on the assumption (or better a demonstration) that connections are not altered during acute slice preparation. While it may be true that large-scale connectivity could not change in a few minutes during slice prep, however it is a shame that the authors do not reference the work of Kristin Harris' group showing that spines are lost on neurons in slices during cold shock (Kirov et al., Neuroscience 2004). There was also a follow-up paper showing that this can be largely avoided by preparing slices at room temp (Bourne et al., Neuropharmacology 2007). In the current manuscript, the authors are slicing before fixation at ice-temp. They do not state so clearly in the Materials and methods, but one has to assume from the Materials and methods description here that the slices then went directly into ice-cold fixative. So there is no warm recovery period and one would expect that there would be a severe loss of spines and other changes in microtubules, known to be cold sensitive. True, the synapses are probably still there, but they would appear to be onto dendrites instead of spines. This could make it difficult to distinguish excitatory and inhibitory synapses, for one thing. It means that this approach would not be well suited for use in any study focused on neuronal morphology, rather than just finding connections (with the concern about artifacts of slice preps as above). The authors should ensure that they further emphasize these potential limitations of their approach.*

We apologize for the omission of the fixative temperature. All tissues were fixed at room temperature in the original manuscript. We have added this information to the Materials and methods section. We have also added example images of tissues that were dissected, vibratome sectioned and fixed at room temperature according to the work from Dr. Harris' group (see Figure 1—figure supplement 2). The results show that ECS volume is preserved equivalently to those sections that were cut in cold buffer. We have added a sentence noting this to the main text (“Preservation of ECS was similar whether tissue was sectioned at 4 °C or 20 °C (Figure 1—figure supplement 2), indicating the described protocol is suitable if alterations of neuronal morphology due to cold shock are a concern (Kirov, Petrak et al. 2004; Bourne, Kirov et al. 2007)”).

We were very explicit in the manuscript that the preservation of in vivo-like morphologies was not the goal of our study. We feel that the results of a recent study in *eLife* by Korogod, Peterson and Knott clearly demonstrate the value of freezing tissue if the goal is to measure in vivo-like neuronal morphologies. We have added this reference to the manuscript (Discussion).

*4) In the improved diffusion depths of antibodies experiment, authors should provide the negative control data in both fixation conditions. Did authors harvest the retina pieces in the same location? Sample size is critical for the antibody penetration because flat mounted retina has different thickness in the central, middle and peripheral portion. Additionally, slice images of XZ, YZ should be added in the Figure 4 to provide convincing idea.*

The samples were all obtained from approximately the same retinal location, near the midpoint between the optic disk and the peripheral edge of the retina. We have added this detail to the Materials and methods section (subsection “Tissue fixation protocols”). The images in Figure 5 are, in fact, cross-sections that were cut prior to imaging. That is, flat mounted retinas were immuno-labeled, embedded in agarose and then cross-sectioned before imaging. The cross-section face is what is presented in Figure 5. This information was already in the Materials and methods section. We updated the Figure 5 legend to indicate that these are cross sectional images that directly show penetration depth following a flat mount labeling.

Regarding negative control data for the antibody experiments, we are using commercially available antibodies that have been described previously in the retina literature. Importantly, the labeling patterns we observed are consistent with the known labeling patterns in the retina. Specifically, vACHT labels the two cholinergic bands in the IPL of the retina and vGAT labels throughout the depth of the IPL. We have noted this in the revision and added references that indicate these are the expected labeling patterns (subsection “Improved diffusion depths of antibodies”). Given that we reproduced the known labeling patterns that numerous other retinal neurobiologists have observed, we do not see the value in repeating negative control experiments.